# Comparison of Single-Port Laparoscopy with Other Surgical Approaches in Endometrial Cancer Surgical Staging: Propensity-Score-Matched Analysis

**DOI:** 10.3390/cancers15225322

**Published:** 2023-11-08

**Authors:** Sang Hyun Cho, Jung-Yun Lee, Eun Ji Nam, Sunghoon Kim, Young Tae Kim, Sang Wun Kim

**Affiliations:** 1Department of Obstetrics and Gynecology, Graduate School, Yonsei University College of Medicine, Seoul 03722, Republic of Korea; francescosh@naver.com; 2Department of Obstetrics and Gynecology, St. Vincent’s Hospital, The Catholic University of Korea, Suwon 16247, Republic of Korea; 3Department of Obstetrics and Gynecology, Women’s Cancer Center, Yonsei Cancer Center, Institute of Women’s Life Medical Science, Yonsei University College of Medicine, Seoul 03722, Republic of Korea; jungyunlee@yuhs.ac (J.-Y.L.); nahmej6@yuhs.ac (E.J.N.); shkim70@yuhs.ac (S.K.); ytkchoi@yuhs.ac (Y.T.K.)

**Keywords:** endometrial cancer, staging surgery, single-port laparoscopy, propensity score matching

## Abstract

**Simple Summary:**

This study compared the long-term surgical outcomes of single-port laparoscopy with other surgical methods (multi-port laparoscopy, robot-assisted laparoscopy, and laparotomy) in endometrial cancer (EC) surgical staging. After conducting propensity score matching, all surgical methods demonstrated comparable survival outcomes with respect to disease-free survival and overall survival. Consequently, single-port laparoscopy is deemed a viable option for surgical staging in EC.

**Abstract:**

This single-institution, retrospective study aimed to compare the surgical outcomes of single-port, multi-port, and robot-assisted laparoscopy, as well as laparotomy, in patients with endometrial cancer who underwent surgical staging between January 2006 and December 2017. This study evaluated various parameters, including disease-free survival (DFS), overall survival (OS), recurrence rate (RR), recurrence site, and intra- and postoperative complications. Propensity score matching was performed to account for baseline characteristics, and a total of 881 patients were included in the analysis. The 3-year DFS of single-port laparoscopy was similar to that of the other groups, but laparotomy exhibited a lower 3-year DFS compared to multi-port (*p* = 0.001) and robot-assisted (*p* = 0.031) laparoscopy. Single-port laparoscopy resulted in a significantly higher 3-year OS than laparotomy (*p* = 0.013). After propensity score matching, the four groups demonstrated similar survival outcomes (3-year DFS: *p* = 0.533; 3-year OS: *p* = 0.328) and recurrence rates (10.3%, 12.1%, 10.3%, and 15.9% in the single-port, multi-port, and robot-assisted laparoscopy and laparotomy groups, respectively, *p* = 0.552). Recurrence most commonly occurred in distant organs. The single-port laparoscopy group had the longest operative time (205.1 ± 76.9 min) but the least blood loss (69.5 ± 90.8 mL) and the shortest postoperative hospital stay (5.2 ± 2.3 days). In contrast, the laparotomy group had the shortest operative time (163.4 ± 51.0 min) but the highest blood loss (368.3 ± 326.4 mL) and the longest postoperative hospital stay (10.3 ± 4.6 days). The transfusion rate was 0% in the single-port laparoscopy group and 3.7% in the laparotomy group. Notably, the laparotomy group had the highest wound complication rate (*p* = 0.001), whereas no wound hernias were observed in the three minimally invasive approaches. In conclusion, the survival outcomes were comparable between the methods, with the benefit of lower blood loss and shorter hospital stay observed in the single-port laparoscopy group. This study suggests that single-port laparoscopy is a feasible approach for endometrial cancer surgical staging.

## 1. Introduction

The incidence of endometrial cancer is increasing [1,2,3,4]. Despite the good prognosis when diagnosed and treated at an early stage [5], patients with high-risk factors and advanced-stage or recurrent disease have a poor prognosis. Optimal tumor resection has an important prognostic role in recurrent or advanced endometrial cancer [6]. Therefore, adequate staging surgery is important for an accurate diagnosis, appropriate surgical treatment, and good prognosis in these patients [1,7,8].

Endometrial cancer staging surgery is traditionally conducted via laparotomy [9]. Owing to the development of laparoscopic instruments and minimally invasive surgical technologies, the survival rate and surgical outcome for endometrial cancer surgical staging via minimally invasive surgery and laparotomy are comparable [9,10,11,12]. Minimally invasive surgery allows for faster patient recovery, reduced bleeding, smaller incision sites, and better pain relief than laparotomy [13,14,15]. Therefore, minimally invasive surgery is typically performed for endometrial cancer staging [16,17,18,19,20,21]. Recently, single-port laparoscopic surgery was adopted to further minimize the morbidity associated with conventional laparoscopy [22]. Single-port laparoscopic surgery is correlated with low operative morbidity, decreased postoperative pain, a shorter recovery period, and superior cosmesis.

However, few studies have compared the survival and recurrence rates according to the staging method, including single-port laparoscopy, in patients with high-risk factors and advanced-stage endometrial cancer who have a poor prognosis [20]. Investigating the survival outcomes, recurrence patterns, surgical outcomes, and complications of different staging surgery methods in patients with various risk factors and stages of endometrial cancer is expected to help physicians select the appropriate staging method based on the patient’s clinical characteristics.

Therefore, this study aimed to compare the survival outcomes as well as perioperative surgical outcomes of single-port laparoscopy, multi-port laparoscopy, robot-assisted laparoscopy, and laparotomy in endometrial cancer surgical staging.

## 2. Materials and Methods

### 2.1. Ethical Statement

This single-institution, retrospective study was conducted according to the principles expressed in the Declaration of Helsinki and was approved by the Institutional Review Board of Severance Hospital (approval number, 4-2021-1017). The requirement for informed consent was waived, as anonymized data were used.

### 2.2. Study Design and Population

The electronic medical records of Severance Hospital (U-severance 3.0) were used to obtain the data of patients with endometrial cancer who underwent surgical staging between January 2006 and December 2017. Patients with a diagnosis of endometrial cancer who underwent surgical staging, including hysterectomy via single-port laparoscopy, multi-port laparoscopy, robot-assisted laparoscopy, or laparotomy, performed by gynecologic oncologists were included. Patients who underwent preoperative treatment, including surgery, chemotherapy, radiotherapy, and/or hormonal therapy, those with a diagnosis of other gynecologic cancer at the time of surgical staging, and those who underwent cooperative surgery with other surgeons during endometrial cancer staging surgery were excluded from this study. 

### 2.3. Treatment

The surgical staging method was selected by the gynecologic oncology surgeon according to the patient’s disease status and baseline characteristics. Staging surgery included hysterectomy and/or bilateral salpingo-oophorectomy and/or pelvic/para-aortic lymph node dissection/sampling.

Adjuvant therapy, including radiotherapy, chemotherapy, and/or hormonal therapy, was administered according to the histopathological results and patient status based on endometrial cancer treatment guidelines, including the National Comprehensive Cancer Network Guideline and the Korean Society of Gynecologic Oncology Guideline, or consultative treatment recommendations.

### 2.4. Data Collection

Demographic, histopathological, and surgical (including complication) data were extracted from the hospital’s electronic medical records. Demographic data included age, body mass index, and history of abdominal surgery.

Histopathological data included International Federation of Gynecology and Obstetrics (FIGO) stage and grade; histological type (endometrioid vs. non-endometrioid); presence of lymphovascular space; myometrial, cervical stromal, or parametrial invasion; pelvic and para-aortic lymph node dissection/sampling; harvested pelvic and para-aortic lymph nodes; and presence of pelvic and para-aortic lymph node metastases.

Surgical data included operative time, estimated blood loss, intraoperative transfusion, and postoperative length of hospitalization. Surgical complications included the intra- and postoperative complications directly related to the surgical staging procedure.

### 2.5. Study Endpoints

The primary endpoints of this study were disease-free survival and overall survival. Disease-free survival was defined as the time from staging surgery to recurrence or last follow-up. Overall survival was defined as the time from staging surgery to death or last follow-up. The secondary endpoints included the recurrence rate, recurrence site, and rate of intra- and postoperative complications.

### 2.6. Propensity Score Matching

The patients’ baseline clinical characteristics were not balanced among the groups. Therefore, to ensure a balanced comparison among the different surgical methods, propensity scores were calculated using patient age; body mass index; FIGO stage and grade; histological type; presence of lymphovascular space; myometrial, cervical stromal, or parametrial invasion; pelvic and para-aortic lymph node dissection/sampling; and presence of pelvic and para-aortic lymph node metastasis. Propensity score matching was performed according to the logistic regression estimation and nearest neighbor matching algorithms with no caliper widths and a 1:1 match between the staging methods (single-port laparoscopy vs. multi-port laparoscopy, single-port laparoscopy vs. robot-assisted laparoscopy, and single-port laparoscopy vs. laparotomy).

### 2.7. Statistical Analysis

All statistical analyses were performed using IBM SPSS Statistics for Windows, version 25.0 (IBM Corp., Armonk, NY, USA). Continuous variables are presented as mean and standard deviation. Continuous variables were analyzed using Student’s *t*-test, one-way analysis of variance, the Mann–Whitney U test, or the Kruskal–Wallis H test. The differences in proportions were analyzed using Fisher’s exact test or the chi squared test. Propensity score matching was performed using the Integration Plug-in for R of SPSS Statistics (IBM Corp.). A standard mean difference of <0.25 was considered adequate after propensity score matching. The recurrence and survival rates were analyzed using Kaplan–Meier survival curves and the log-rank test. Statistical significance was set at *p* < 0.05.

## 3. Results

### 3.1. Initial Analysis

#### 3.1.1. Patient Characteristics

The patients’ baseline characteristics are shown in Appendix A. A total of 881 patients were included and divided into four groups according to the staging method: single-port laparoscopy (*n* = 107; 12.1%), multi-port laparoscopy (*n* = 299; 33.9%), robot-assisted laparoscopy (*n* = 207; 30.4%), and laparotomy (*n* = 268; 23.5%). The median age and body mass index were 54.1 years and 24.8 kg/m^2^, respectively, without significant differences among the groups (*p* = 0.131 and *p* = 0.663, respectively).

The histopathological factors related to poor prognosis (FIGO stage and grade; histological type; presence of lymphovascular space; myometrial, cervical stromal, or parametrial invasion; and presence of pelvic and para-aortic lymph node metastasis) were significantly different between the groups (all *p* < 0.001). Across all groups, endometrioid endometrial cancer was the most common histological type (single-port laparoscopy: *n* = 90, 84.1%; multi-port laparoscopy: *n* = 271, 90.6%; robot-assisted laparoscopy: *n* = 184, 88.5%; and laparotomy: *n* = 187, 69.8%), and FIGO stage IA and FIGO differentiation grade I were the most common staging findings.

#### 3.1.2. Surgical Outcomes before Propensity Score Matching

The perioperative surgical outcomes are presented in Table 1. The operative time was longest in the single-port laparoscopy group (205.1 ± 76.5 min) and shortest in the multi-port laparoscopy group (159.2 ± 64.1 min). The estimated blood loss was lowest in the single-port laparoscopy group (69.5 ± 90.8 mL) and highest in the laparotomy group (421.0 ± 442.8 mL). The intraoperative transfusion rate was 0% in the single-port laparoscopy group and 6.7% in the laparotomy group. The postoperative length of hospitalization was shortest in the single-port laparoscopy group (5.2 ± 2.3 days). The number of harvested para-aortic lymph nodes was significantly higher in the single-port laparoscopy group (14.1 ± 10.7) than in the other staging method groups (*p* < 0.001).

Radiotherapy, chemotherapy, and/or hormonal therapy were performed for adjuvant treatment in 226 (25.7%), 189 (21.5%), and 8 (0.9%) patients, respectively. More than half of the patients (*n* = 522, 59.3%) did not undergo adjuvant treatment postoperatively (Table 1). The adjuvant therapy rate, especially the chemotherapy rate, was higher in the single-port laparoscopy and laparotomy groups than in the other groups (all *p* < 0.001).

#### 3.1.3. Survival Outcomes before Propensity Score Matching

Before propensity score matching, the follow-up durations were 51.5 ± 18.2, 66.7 ± 33.8, 67.2 ± 35.5, and 65.8 ± 42.7 months in the single-port laparoscopy, multi-port laparoscopy, robot-assisted laparoscopy, and laparotomy groups, respectively. The 3-year disease-free survival rate of the single-port laparoscopy group (90.4%) was not significantly different than that of the multi-port laparoscopy (92.3%, *p* = 0.711), robot-assisted laparoscopy (92.0%, *p* = 0.825), or laparotomy (83.9%, *p* = 0.063) group. However, the 3-year disease-free survival rate of the laparotomy group (83.9%) was significantly lower than that of the multi-port laparoscopy (92.3%, *p* = 0.001) and robot-assisted laparoscopy (92.0%, *p* = 0.013) groups (Table 2 and Figure 1). The 3-year overall survival rates were also significantly lower in the laparotomy group than in the single-port laparoscopy (*p* = 0.013), multi-port laparoscopy (*p* = 0.030), and robot-assisted laparoscopy (*p* = 0.027) groups, although there were no significant differences between the single-port laparoscopy group and the multi-port laparoscopy (*p* = 0.224) or robot-assisted laparoscopy (*p* = 0.360) groups.

When the patients were divided according to the FIGO stage, the 3-year disease-free survival and overall survival rates were not different between the treatment groups for each FIGO stage group (disease-free survival: stage I, *p* = 0.417; stage II, *p* = 0.610; stages III and IV, *p* = 0.167; overall survival: stage I, *p* = 0.658; stage II, *p* = 0.212; stages III and IV, *p* = 0.092; Table 2 and Figure 2). The 3-year disease-free survival rates of single-port laparoscopy were 92.9%, 100%, and 71.4% among patients with FIGO stage I, II, and III and IV endometrial cancer, respectively, and were not significantly different to those of other surgical staging methods.

The single-port laparoscopy group had the highest 3-year overall survival rates for each FIGO stage group (stage I, 98.8%; stage II, 100; and stages III and IV, 100%), although they were not significantly different from those of the other staging methods.

The overall recurrence rate was 12.5% (110/881), with a significant difference between the groups (single-port laparoscopy, 10.3%; multi-port laparoscopy, 9.4%; robot-assisted laparoscopy, 10.6%; and laparotomy, 18.3%; *p* = 0.007). The most common sites of recurrence were the distant organs, including the lungs, pleura, bones, brain, and adrenal glands (*n* = 39, 4.4%), and the abdomen (*n* = 21, 2.4%). Abdominal recurrence was significantly different between the staging method groups (*p* = 0.04; Table 2).

### 3.2. Propensity-Score-Matched Analysis

#### 3.2.1. Patient Characteristics after Propensity Score Matching

A total of 428 patients (107 from each group) were included in the propensity score matching. After matching, the baseline characteristics of the patients were not significantly different between the groups (Appendix A).

#### 3.2.2. Surgical Outcomes after Propensity Score Matching

After propensity score matching, the single-port laparoscopy group had the longest operative time (205.1 ± 76.9 min), and the laparotomy group had the shortest operative time (163.4 ± 51.0 min) (Appendix A). The estimated blood loss was the lowest in the single-port laparoscopy group (69.5 ± 90.8 mL) and highest in the laparotomy group (368.3 ± 326.4 mL). The transfusion rate was 0% in the single-port laparoscopy group and 3.7% in the laparotomy group. The single-port laparoscopy group had the shortest postoperative hospital stay (5.2 ± 2.3 days), while the laparotomy group had the longest postoperative hospital stay (10.3 ± 4.9 days). The number of harvested pelvic lymph nodes was the greatest in the laparotomy group but was not significantly different between the three minimally invasive surgery groups. The number of harvested para-aortic lymph nodes was the largest in the single-port laparoscopy group. There were no significant differences between the groups regarding the adjuvant treatment modalities or the lack of adjuvant treatment.

#### 3.2.3. Survival Outcomes after Propensity Score Matching

After propensity score matching, the mean follow-up duration was shortest in the single-port laparoscopy group, followed by the multi-port laparoscopy group, laparotomy group, and robot-assisted laparoscopy group, with a preserved statistically significant difference between the groups (*p* = 0.004).

The disease-free survival and overall survival were not significantly different between the surgical staging method groups after propensity score matching (*p* = 0.533 and *p* = 0.328, respectively; Table 3 and Figure 1).

Similarly, the recurrence rates were not significantly different between the surgical staging method groups (*p* = 0.552). The recurrence rates were lowest in the single-port laparoscopy and robot-assisted laparoscopy groups and highest in the laparotomy group (10.3%, 10.3%, and 15.9%, respectively; *p* = 0.552). The most common sites of recurrence were the distant organs and were not significantly different between the groups.

#### 3.2.4. Complications

The intra- and postoperative complications after propensity score matching are summarized in Appendix A. Notably, wound problems were the most prevalent postoperative complications, observed exclusively in the laparotomy group (5.6%, *p* = 0.001). Importantly, there were no significant differences in intraoperative and postoperative complications among the four surgical methods, with the exception of wound problems. Furthermore, it is worth highlighting that minimally invasive approaches, including single-port laparoscopy, exhibited no cases of incisional hernia.

## 4. Discussion

After adjusting for baseline characteristics through propensity score matching, there were no significant differences in survival rates and safety outcomes among the various surgical staging methods for endometrial cancer. This included patients at all FIGO stages and with different histological types. Notably, no patients experienced severe surgical complications.

Several studies comparing minimally invasive surgery with open surgery have consistently reported similar surgical outcomes [9,10,11,17,18,20,21,23,24,25,26,27,28]. A specific study comparing various minimally invasive surgery methods (single-port laparoscopy, multi-port laparoscopy, and robot-assisted laparoscopy) found no significant differences in survival outcomes among patients with stage IA/IB endometrial cancer [18]. These findings underline that patients’ prognoses do not differ when the surgery is performed by highly skilled surgeons. Furthermore, another study reported no significant differences in survival outcomes for patients with high-risk endometrial cancer who underwent laparoscopy or robot-assisted laparoscopy [29]. However, our study is the first to compare outcomes across various minimally invasive surgical approaches (single-port laparoscopy, multi-port laparoscopy, and robot-assisted laparoscopy) and open surgery in patients with high-risk endometrial cancer. After propensity score matching, we found no significant differences in survival outcomes between single-port laparoscopy and other surgical staging methods or among patients with high-risk factors.

In the Gynecologic Oncology Group LAP2 study comparing laparoscopy with laparotomy for uterine cancer staging, laparoscopy had a longer median operative time (*p* < 0.001), and intraoperative complications included vein, bowel, artery, bladder, and ureter injuries, with similar rates between the two methods (*p* = 0.106) [9]. However, the rate of postoperative adverse events (grade ≥ 2) significantly differed between the methods (*p* < 0.001), with ileus, fever, wound infection, urinary tract infection, arrhythmia, pneumonia, pulmonary embolus, venous thrombophlebitis, bowel obstruction, and congestive heart failure being the most common complications. In our study, after propensity score matching, the single-port laparoscopy group exhibited the longest operative time and the shortest hospital stay. Conversely, the laparotomy group had the shortest operative time but the longest hospital stay. The single-port laparoscopy group may have required more time due to the increased number of harvested para-aortic lymph nodes. Also, it is noteworthy that single-port laparoscopy resulted in the least amount of bleeding, and none of the patients in this group required intraoperative transfusion.

Minimally invasive approaches, including single-port laparoscopy, exhibited a complete absence of wound-related problems, including incisional hernias, throughout the entire 3-year long-term follow-up period. Conversely, wound complications were exclusively observed in the laparotomy group. Therefore, it is essential to acknowledge that wound-related complications are more common after open surgery. Thus, it becomes crucial to select the most appropriate surgical method while considering pre- and postoperative treatments, especially for patients at a higher risk of experiencing such complications.

Endometrial cancer recurrence can occur in various sites, including the vagina, pelvic and para-aortic lymph nodes, peritoneum, and lungs, but is less common in extra-abdominal nodes, intra-abdominal organs, musculoskeletal and soft tissues, and the central nervous system [30]. In the Gynecologic Oncology Group LAP2 Study, recurrence sites included the lungs, multiple sites, vagina, abdomen, pelvis, lymph nodes, liver, and bones, and recurrence often occurred in multiple sites, with no significant differences reported between the groups (*p* = 0.470) [31]. In our study, distant organs, including the lungs, were the most common sites of recurrence, both before and after propensity score matching. The abdominal recurrence rates significantly differed between the groups before propensity score matching. The recurrence patterns and frequencies observed in our study, based on the surgical staging method, align with those reported in previous studies. However, since postoperative adjuvant therapy can influence recurrence, future studies should explore the association between recurrence site and surgical method, while also considering the effect of postoperative adjuvant therapy.

Our study included two patients with abdominal recurrence: one with an abdominal mass in the laparotomy group and one with an abdominal wall mass in the robot-assisted laparoscopy group. However, we did not confirm whether the recurrence sites corresponded to the surgical incision or trocar sites. In the Gynecologic Oncology Group LAP2 study, four patients had possible trocar site recurrence, including three with advanced-stage endometrial cancer [31]. In contrast, trocar site recurrence was not reported in patients with early-stage endometrial cancer [26]. This suggests that the risk of trocar site recurrence may be higher in patients with advanced-stage endometrial cancer. Further studies should compare incision site recurrence with trocar site recurrence and investigate the impact of incision/trocar site recurrence on survival rates.

Previous studies have explored the feasibility and prognosis of various surgical staging methods, but no simultaneous comparison of single-port laparoscopy, multi-port laparoscopy, robotic surgery, and laparotomy in prospective studies has been conducted. In our study, propensity score matching was applied to mitigate issues related to retrospective data-based research. Nevertheless, our study remains retrospective, potentially introducing selection bias.

One potential source of bias in our study lies in the selection of surgical methods. The choice of method might vary due to advancements in surgical techniques and individual patient factors such as health status and costs. Surgeon preference and experience also significantly influence the selection of the appropriate technique. Our analysis revealed significant differences in recurrence rates among the surgical methods before propensity score matching. The laparotomy group had a higher prevalence of patients with advanced stage or non-endometrioid histology, representing high-risk groups, potentially contributing to the higher recurrence rates in this group. After propensity score matching, while overall recurrence rates equalized across the surgical methods, laparotomy continued to exhibit a higher recurrence rate compared to the other methods. This finding suggests that patients with more extensive disease or challenging clinical conditions, making them less suitable candidates for minimally invasive approaches, might have been preferentially directed towards laparotomy.

Secondly, pelvic and para-aortic lymph node dissection/sampling in the surgical staging of endometrial cancer serves diagnostic and therapeutic purposes. Traditionally, systemic lymph node dissection was performed, leading to complications such as lymphedema, bleeding, and nerve damage. To enhance the safety of lymph node assessment and accuracy of surgical staging, sentinel lymph node biopsy has been increasingly adopted [32,33,34]. Two methods, the one-step and the two-step method, are commonly used in sentinel lymph node biopsy, with the latter enhancing accuracy, especially in para-aortic lymph node assessment [35,36]. In our study, both before and after propensity score matching, significant differences in harvested pelvic and para-aortic lymph nodes were observed among the surgical groups. Particularly, the laparotomy group, where systemic lymph node dissection was prevalent, exhibited higher numbers of harvested pelvic lymph nodes. In some cases within the single-port laparoscopy group, the use of the two-step method led to increased para-aortic lymph node harvest compared to other groups. These differences in lymph node dissection techniques might introduce biases into the study results.

Additionally, postoperative treatments significantly influence patient outcomes. While guidelines for endometrial cancer treatments provide recommendations based on previous studies, the actual administration of adjuvant therapy can vary based on patient condition and institutional protocols [1,7,37,38,39,40,41]. Before propensity score matching, we observed significant differences in postoperative adjuvant treatment patterns among the surgical groups. Chemotherapy and radiotherapy were more frequently administered in patients undergoing laparotomy or single-port laparoscopy, reflecting a higher prevalence of high-risk factors like PALN metastasis, advanced stage, or non-endometrioid histology in these groups. However, after propensity score matching, the differences in adjuvant treatment became non-significant, indicating a balanced postoperative management among the groups.

While propensity score matching helped mitigate biases, the inherent challenges in retrospective analyses persist. To achieve more precise and conclusive results, well-designed prospective studies are crucial. It is essential to aim at minimizing and overcoming the biases previously discussed, as this could lead to better treatments for patients.

Our study had several strengths. First, we analyzed survival and recurrence rates across various endometrial cancer stages, histological types, and risk factors for poor prognosis in response to four surgical staging methods. Second, we employed propensity score matching to eliminate baseline characteristic differences among the groups. As a result, this study’s outcomes can be considered similar to those of a prospective study.

However, there were certain limitations to our study. First, it was not feasible to compare all four groups simultaneously due to a statistical limitation of propensity score matching. Second, the selection of surgical methods was made at the time of care, introducing potential bias. Third, our study did not assess postoperative pain, recovery, or quality of life. Finally, fewer patients with FIGO stage IV endometrial cancer underwent minimally invasive surgery compared to laparotomy.

## 5. Conclusions

After propensity score matching, single-port laparoscopy yielded comparable outcomes to laparotomy, multi-port laparoscopic surgery, and robot-assisted surgery for endometrial cancer surgical staging. This real-world data-based study confirms that well-trained, single-port laparoscopic gynecologic surgeons can successfully perform single-port laparoscopy in endometrial cancer patients with diverse stages and histology. Future prospective studies should further investigate the influence of different surgical staging methods on endometrial cancer, considering various risk factors and stages.

## Figures and Tables

**Figure 1 cancers-15-05322-f001:**
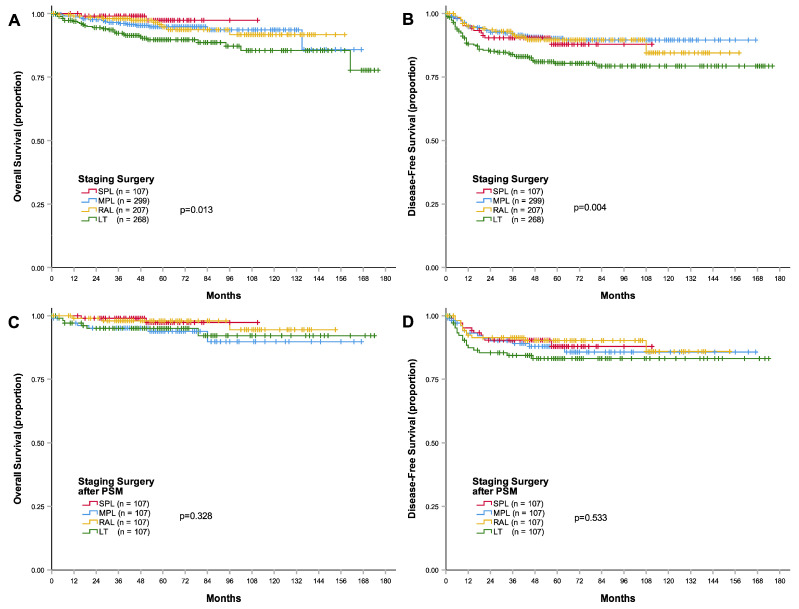
Kaplan–Meier curves of survival outcomes according to the surgical methods in endometrial cancer. (**A**) Overall survival curves before matching, (**B**) disease-free survival curves before matching, (**C**) overall survival curves after matching, and (**D**) disease-free survival curves after matching. Abbreviations: LT, laparotomy; MPL, multi-port laparoscopy; RAL, robot-assisted laparoscopy; SPL, single-port laparoscopy; PSM, propensity score matching.

**Figure 2 cancers-15-05322-f002:**
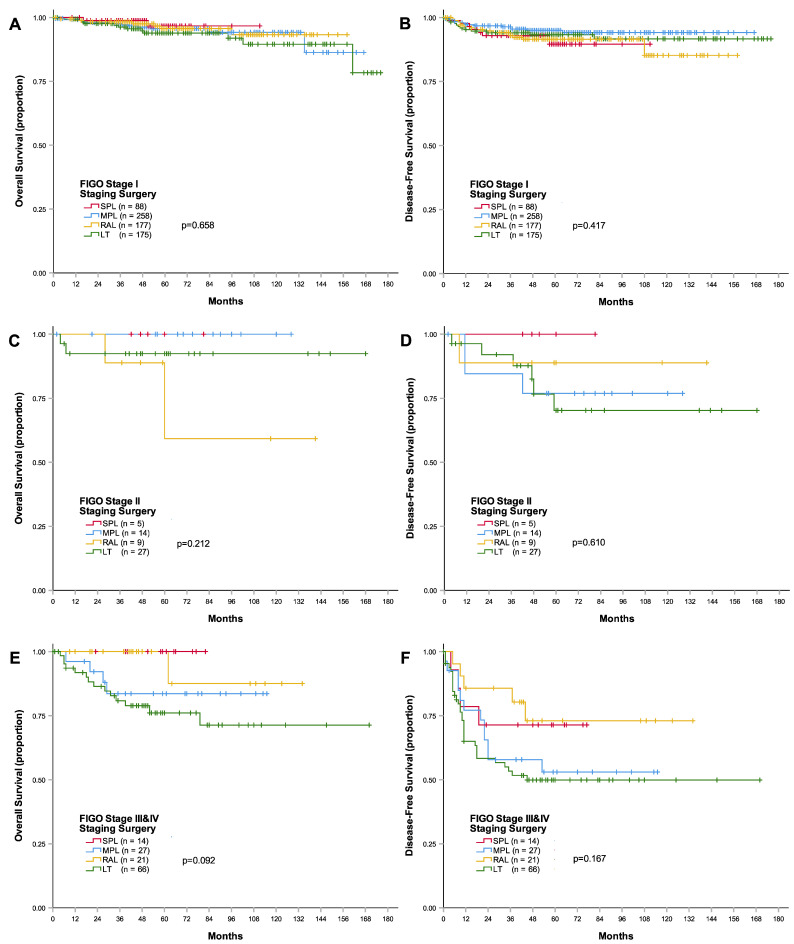
Kaplan–Meier curves of survival outcomes according to the FIGO stage. (**A**) Overall survival curves of patients with FIGO stage I endometrial cancer, (**B**) disease-free survival of patients with FIGO stage I endometrial cancer, (**C**) overall survival of patients with FIGO stage II endometrial cancer, (**D**) disease-free survival of patients with FIGO stage II endometrial cancer, (**E**) overall survival of patients with FIGO stages III and IV endometrial cancer, and (**F**) disease-free survival of patients with FIGO stages III and IV endometrial cancer. Abbreviations: FIGO, International Federation of Gynecology and Obstetrics; LT, laparotomy; MPL, multi-port laparoscopy; RAL, robot-assisted laparoscopy; SPL, single-port laparoscopy.

**Table 1 cancers-15-05322-t001:** Perioperative surgical outcomes and adjuvant treatments of four different surgical staging methods for endometrial cancer (*n* = 881).

	Total (*n* = 881)	SPL (*n* = 107)	MPL (*n* = 299)	RAL (*n* = 207)	LT (*n* = 268)	*p*-Value
Operative time, mean ± SD, min	173.0 ± 66.9	205.1 ± 76.9	159.2 ± 64.1	178.8 ± 57.9	171.0 ± 67.6	<0.001
Estimated blood loss, mean ± SD, mL	212.4 ± 305.4	69.5 ± 90.8	143.4 ± 167.0	115.9 ± 127.6	421.0 ± 442.8	<0.001
Intraoperative transfusion, *n* (%)	24 (2.7)	0	4 (1.3)	2 (1.0)	18 (6.7)	<0.001
Postoperative length of hospitalization, mean ± SD, days	8.1 ± 3.9	5.2 ± 2.3	7.5 ± 3.4	7.2 ± 2.4	10.7 ± 4.5	<0.001
Harvested PLNs, mean ± SD, *n*	17.3 ± 9.5	16.1 ± 9.5	17.0 ± 9.4	15.9 ± 9.2	19.4 ± 9.5	<0.001
Harvested PALNs, mean ± SD, *n*	7.4 ± 7.9	14.1 ± 10.7	7.0 ± 7.0	5.8 ± 5.7	6.3 ± 7.5	<0.001
Adjuvant therapy, *n* (%)						
Radiotherapy	226 (25.7)	24 (22.4)	58 (19.4)	42 (20.3)	102 (38.1)	<0.001
Chemotherapy	189 (21.5)	32 (29.9)	40 (13.4)	32 (15.5)	85 (31.7)	<0.001
Hormonal therapy	8 (0.9)	3 (2.8)	1 (0.3)	1 (0.5)	3 (1.1)	0.136
None	522 (59.3)	58 (54.2)	214 (71.6)	140 (67.6)	110 (41.0)	<0.001

Abbreviations: LT, laparotomy; MPL, multi-port laparoscopy; PALN, para-aortic lymph node; PLN, pelvic lymph node; RAL, robot-assisted laparoscopy; SD, standard deviation SPL, single-port laparoscopy.

**Table 2 cancers-15-05322-t002:** Survival outcomes and recurrence sites according to the surgical staging methods in endometrial cancer (*n* = 881).

Parameters	Total (*n* = 881)	SPL (*n* = 107)	MPL (*n* = 299)	RAL (*n* = 207)	LT (*n* = 268)	*p*-Value
Duration of follow-up, mean ± SD, months	64.7 ± 36.1	51.5 ± 18.2	66.7 ± 33.8	67.2 ± 35.5	65.8 ± 42.7	0.001
Disease-free survival, % *						0.004
3-year	89.4	90.4	92.3	92.0	83.9	
5-year	86.9	87.9	90.3	89.6	80.4	
Overall survival, %*						0.013
3-year	95.9	99	96.5	98	92.3	
5-year	93.6	97.4	94.8	94.7	89.7	
3-year disease-free survival, % *						
FIGO stage I	94.5	92.9	96.4	92.9	94.1	0.417
FIGO stage II	90.3	100	84.6	88.9	92.1	0.610
FIGO stages III and IV	61.6	71.4	57.9	85.7	53.4	0.167
3-year overall survival, % *						
FIGO stage I	97.6	98.8	97.6	98.3	96.3	0.658
FIGO stage II	94.3	100	100	88.9	92.4	0.212
FIGO stages III and IV	86.8	100	83.6	100	80.9	0.092
Recurrence, *n* (%)	110 (12.5)	11 (10.3)	28 (9.4)	22 (10.6)	49 (18.3)	0.007
Recurrence site, *n* (%) **						
Vaginal vault	16 (1.8)	1 (0.9)	5 (1.7)	3 (1.4)	7 (2.6)	0.757
Pelvis	19 (2.2)	1 (0.9)	8 (2.7)	5 (2.4)	5 (1.9)	0.785
Abdomen	21 (2.4)	3 (2.8)	3 (1.0)	3 (1.4)	12 (4.5)	0.040
Peritoneum	13 (1.5)	1 (0.9)	4 (1.3)	1 (0.5)	7 (2.6)	0.301
Distant organ	39 (4.4)	4 (3.7)	12 (4.0)	9 (4.3)	14 (5.2)	0.886
PLN	19 (2.2)	1 (0.9)	4 (1.3)	5 (2.4)	9 (3.4)	0.361
PALN	15 (1.7)	3 (2.8)	5 (1.7)	1 (0.5)	6 (2.2)	0.311
Supradiaphragmatic LN	9 (1.0)	0	4 (1.3)	0	5 (1.9)	0.157
Inguinal LN	2 (0.2)	0	0	1 (0.5)	1 (0.4)	0.634

* Kaplan–Meier estimate, ** each site of recurrence in a patient was counted individually. Abbreviations: FIGO, International Federation of Gynecology and Obstetrics; LN, lymph node; LT, laparotomy; MPL, multi-port laparoscopy; PALN, para-aortic LN; PLN, pelvic LN; RAL, robot-assisted laparoscopy; SD, standard deviation; SPL, single-port laparoscopy.

**Table 3 cancers-15-05322-t003:** Propensity-score-matched analysis of survival outcomes and recurrence sites according to the surgical staging methods in endometrial cancer (*n* = 428).

Parameters	SPL (*n* = 107)	MPL (*n* = 107)	RAL (*n* = 107)	LT (*n* = 107)	*p*-Value
Duration of follow-up, mean ± SD, months	51.5 ± 18.2	64.1 ± 33.9	68.4 ± 36.9	66.3 ± 40.6	0.004
Disease-free survival, % *					0.533
3-year	90.4	90.1	91.4	84.4	
5-year	87.9	87.9	90.2	83.2	
Overall survival, % *					0.328
3-year	99.0	95.2	98.0	95.1	
5-year	97.4	93.9	98.0	95.1	
Recurrence, *n* (%)	11 (10.3)	13 (12.1)	11 (10.3)	17 (15.9)	0.552
Recurrence site, *n* (%) **					
Vaginal vault	1 (0.9)	3 (2.8)	2 (1.9)	0	0.525
Pelvis	1 (0.9)	2 (1.9)	1 (0.9)	0	0.905
Abdomen	3 (2.8)	1 (0.9)	1 (0.9)	4 (3.7)	0.472
Peritoneum	1 (0.9)	0	0	3 (2.8)	0.200
Distant organ	4 (3.7)	5 (4.7)	5 (4.7)	8 (7.5)	0.728
PLN	1 (0.9)	1 (0.9)	4 (3.7)	4 (3.7)	0.335
PALN	3 (2.8)	3 (2.8)	1 (0.9)	3 (2.8)	0.778
Supradiaphragmatic LN	0	1 (0.9)	0	1 (0.9)	1.000
Inguinal LN	0	0	1 (0.9)	1 (0.9)	1.000

* Kaplan–Meier estimate, ** each site of recurrence in a patient was counted individually. Abbreviations: LN, lymph node; LT, laparotomy; MPL, multi-port laparoscopy; PALN, para-aortic LN; PLN, pelvic LN; RAL, robot-assisted laparoscopy; SD, standard deviation; SPL, single-port laparoscopy.

## Data Availability

Data are available from the corresponding author upon reasonable request.

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
