# Peer review of "Comparison of Single-Port Laparoscopy with Other Surgical Approaches in Endometrial Cancer Surgical Staging: Propensity-Score-Matched Analysis"

_cancers, 2023, doi:10.3390/cancers15225322_

Round 1
Reviewer 1 Report
Comments and Suggestions for Authors
Interesting data. Needs extensive discussion regarding selection bias
Comments on the Quality of English LanguageGood
Author Response
Thank you for the important and valuable comments on our manuscript. We have revised our manuscript according to the reviewer’s comments, as follows.
Reviewer's comments:
Interesting data. Needs extensive discussion regarding selection bias:
→Response: Thank you for the valuable comment. We have incorporated discussions on selection bias, the second reviewer's comments, and common aspects into the discussion section. Considering factors that could introduce bias, we addressed the following aspects: firstly, bias related to the choice of surgical methods; secondly, differences in lymph node dissection techniques; and thirdly, variations in postoperative adjuvant therapy. We described these factors as follows:
“Previous studies have explored the feasibility and prognosis of various surgical staging methods, but no simultaneous comparison of single-port laparoscopy, multi-port laparoscopy, robotic surgery, and laparotomy in prospective studies has been conducted. In our study, propensity score matching was applied to mitigate issues related to retrospective data-based research. Nevertheless, our study remains retrospective, potentially introducing selection bias.
One potential source of bias in our study lies in the selection of surgical methods. The choice of method might vary due to advancements in surgical techniques and individual patient factors such as health status and costs. Surgeon preference and experience also significantly influence the selection of the appropriate technique. Our analysis revealed significant differences in recurrence rates among the surgical methods before propensity score matching. The laparotomy group had a higher prevalence of patients with advanced stage or non-endometrioid histology, representing high-risk groups, potentially contributing to the higher recurrence rates in this group. After propensity score matching, while overall recurrence rates equalized across surgical methods, laparotomy continued to exhibit a higher recurrence rate compared to other methods. This finding suggests that patients with more extensive disease or challenging clinical conditions, making them less suitable candidates for minimally invasive approaches, might have been preferentially directed towards laparotomy.
Secondly, pelvic and para-aortic lymph node dissection/sampling in the surgical staging of endometrial cancer serves diagnostic and therapeutic purposes. Traditionally, systemic lymph node dissection was performed, leading to complications such as lymphedema, bleeding, and nerve damage. To enhance the safety of lymph node assessment and accuracy of surgical staging, sentinel lymph node biopsy has been increasingly adopted [32-34]. Two methods, the one-step and the two-step method, are commonly used in sentinel lymph node biopsy, with the latter enhancing accuracy, especially in para-aortic lymph node assessment [35,36]. In our study, both before and after propensity score matching, significant differences in harvested pelvic and para-aortic lymph nodes were observed among the surgical groups. Particularly, the laparotomy group, where systemic lymph node dissection was prevalent, exhibited higher numbers of harvested pelvic lymph nodes. In some cases within the single-port laparoscopy group, the use of the two-step method led to increased para-aortic lymph node harvest compared to other groups. These differences in lymph node dissection techniques might introduce biases in-to the study results.
Additionally, postoperative treatments significantly influence patient outcomes. While guidelines for endometrial cancer treatments provide recommendations based on previous studies, the actual administration of adjuvant therapy can vary based on patient condition and institutional protocols [1,7,37-41]. Before propensity score matching, we observed significant differences in postoperative adjuvant treatment patterns among the surgical groups. Chemotherapy and radiotherapy were more frequently administered in patients undergoing laparotomy or single-port laparoscopy, reflecting a higher prevalence of high-risk factors like PALN metastasis, advanced stage, or non-endometrioid histology in these groups. However, after propensity score matching, the differences in adjuvant treatment became non-significant, indicating a balanced postoperative management among the groups.
While propensity score matching helped mitigate biases, the inherent challenges in retrospective analyses persist. To achieve more precise and conclusive results, well-designed prospective studies are crucial. It is essential to aim at minimizing and overcoming the biases previously discussed, as this could lead to better treatments for patients.”
Reviewer 2 Report
Comments and Suggestions for Authors
In line 167-169 ‘The number of harvested para-aortic lymph nodes was significantly higher in the single-port laparoscopy’. Is there any comment of this result? May it be related to patient selection ( less BMI in single port method and etc) ?.
In 173-174, Adjuvant chemotherapy rate is higher in the single-port laparoscopy and laparotomy groups. This data may affect oncological results.
In line 205-207, most common recurrence was seen in laparotomy group. It s nearly 2 times more in laparotomy group than single port group. A command is it necessary.
In figure 1A, Y axis should be DFS, in 1B axis should be OS, 1C should be DFS and ID should be OS. Please check.
In figure 2, there are some similar problem in definition of Y axis.
Author Response
Thank you for the important and valuable comments on our manuscript. We have revised our manuscript according to the reviewer’s comments, as follows.
Reviewer's comments:
In line 167-169 ‘The number of harvested para-aortic lymph nodes was significantly higher in the single-port laparoscopy’. Is there any comment of this result? May it be related to patient selection (less BMI in single port method and etc) ?.
→Response: Thank you for pointing out the shortcomings. We commented the result of harvested para-aortic lymph nodes in the single-port laparoscopy in discussion. Two-step method of sentinel biopsy was adopted in some patients for single-port laparoscopy. This method was led to increased para-aortic lymph node harvest compared to other groups.
“Secondly, pelvic and para-aortic lymph node dissection/sampling in the surgical staging of endometrial cancer serves diagnostic and therapeutic purposes. Traditionally, systemic lymph node dissection was performed, leading to complications such as lymphedema, bleeding, and nerve damage. To enhance the safety of lymph node assessment and accuracy of surgical staging, sentinel lymph node biopsy has been increasingly adopted [32-34]. Two methods, the one-step and the two-step method, are commonly used in sentinel lymph node biopsy, with the latter enhancing accuracy, especially in para-aortic lymph node assessment [35,36]. In our study, both before and after propensity score matching, significant differences in harvested pelvic and para-aortic lymph nodes were observed among the surgical groups. Particularly, the laparotomy group, where systemic lymph node dissection was prevalent, exhibited higher numbers of harvested pelvic lymph nodes. In some cases within the single-port laparoscopy group, the use of the two-step method led to increased para-aortic lymph node harvest compared to other groups. These differences in lymph node dissection techniques might introduce biases in-to the study results.”
In 173-174, Adjuvant chemotherapy rate is higher in the single-port laparoscopy and laparotomy groups. This data may affect oncological results.
→Response: Thank you for pointing out the shortcomings. We discussed about it and descripted in the “Discussion”. In single port laparoscopy and laparoscopic groups, the patients with high-risk factors like PALN metastasis, advanced stage, or non-endometrioid histology were more than other groups. After propensity score matching, there was not significance in the adjuvant treatment among groups.
” Additionally, postoperative treatments significantly influence patient outcomes. While guidelines for endometrial cancer treatments provide recommendations based on previous studies, the actual administration of adjuvant therapy can vary based on patient condition and institutional protocols [1,7,37-41]. Before propensity score matching, we observed significant differences in postoperative adjuvant treatment patterns among the surgical groups. Chemotherapy and radiotherapy were more frequently administered in patients undergoing laparotomy or single-port laparoscopy, reflecting a higher prevalence of high-risk factors like PALN metastasis, advanced stage, or non-endometrioid histology in these groups. However, after propensity score matching, the differences in adjuvant treatment became non-significant, indicating a balanced postoperative management among the groups.”
In line 205-207, most common recurrence was seen in laparotomy group. It’s nearly 2 times more in laparotomy group than single port group. A command is it necessary.
→Response: Thank you for the valuable comment. We descripted in the “Discussion”. The laparotomy group had a higher prevalence of patients with advanced stage or non-endometrioid histology, representing high-risk groups, potentially contributing to the higher recurrence rates in this group. And This finding suggests that patients with more extensive disease or challenging clinical conditions, making them less suitable candidates for minimally invasive approaches, might have been preferentially directed towards laparotomy.
“One potential source of bias in our study lies in the selection of surgical methods. The choice of method might vary due to advancements in surgical techniques and individual patient factors such as health status and costs. Surgeon preference and experience also significantly influence the selection of the appropriate technique. Our analysis revealed significant differences in recurrence rates among the surgical methods before propensity score matching. The laparotomy group had a higher prevalence of patients with advanced stage or non-endometrioid histology, representing high-risk groups, potentially contributing to the higher recurrence rates in this group. After propensity score matching, while overall recurrence rates equalized across surgical methods, laparotomy continued to exhibit a higher recurrence rate compared to other methods. This finding suggests that patients with more extensive disease or challenging clinical conditions, making them less suitable candidates for minimally invasive approaches, might have been preferentially directed towards laparotomy.”
In figure 1A, Y axis should be DFS, in 1B axis should be OS, 1C should be DFS and ID should be OS. Please check.
→Response: Thank you for pointing out the shortcomings. We have revised the legend for figure 1. The errors in DFS and OS have been corrected.
“Figure 1. Kaplan–Meier curves of survival outcomes according to the surgical methods in endometrial cancer; (A) Overall survival curves before matching, (B) disease-free survival curves before matching, (C) overall survival curves after matching, and (D) disease-free survival curves after matching. Abbreviations: LT, laparotomy; MPL, multi-port laparoscopy; RAL, robot-assisted laparoscopy; SPL, single-port laparoscopy; PSM, propensity score matching.”
In figure 2, there are some similar problem in definition of Y axis.
→Response: Thank you for pointing out the shortcomings. We have also revised the legend for Figure 2. The errors in DFS and OS have been corrected.
“Figure 2. Kaplan–Meier curves of survival outcomes according to the FIGO stage: (A) Overall survival curves of patients with FIGO stage I endometrial cancer, (B) disease-free survival of patients with FIGO stage I endometrial cancer, (C) overall survival of patients with FIGO stage II endometrial cancer, (D) disease-free survival of patients with FIGO stage II endometrial cancer, (E) overall survival of patients with FIGO stage III & IV endometrial cancer, and (F) disease-free survival of patients with FIGO stage III & IV endometrial cancer. Abbreviations: FIGO, International Federation of Gynecology and Obstetrics; LT, laparotomy; MPL, multi-port laparoscopy; RAL, robot-assisted laparoscopy; SPL, single-port laparoscopy.”
Reviewer 3 Report
Comments and Suggestions for Authors
The Authors compare the results of endometrial cancer treatment in four ways: by single-port laparoscopy, traditional multi-port laparoscopy, robotic surgery and by laparotomy. The manuscript describes a single-institution, retrospective study in a representative number of 881 patients.
Minimally invasive surgery, as compared to laparotomy, resulted in less blood loss and less wound infections occured, but more operative time. Endoscopic approach evoked less urinary bladder dysfunction.
However, the most important finding is that all four groups of patients demonstrated similar 3-year survival outcomes. This very awaited study provides a sound basis to promote minimally invasive approach in the treatment of endometrial cancer, as opposed to widely discussed data from studies on laparoscopic treatment of cervical cancer.
I highly recommend publishing this paper in the present form to the interest of readers.
Author Response
Thank you for the important and valuable comments on our manuscript. We have revised our manuscript according to the reviewer’s comments, as follows.
Reviewer's comments:
The Authors compare the results of endometrial cancer treatment in four ways: by single-port laparoscopy, traditional multi-port laparoscopy, robotic surgery and by laparotomy. The manuscript describes a single-institution, retrospective study in a representative number of 881 patients.
Minimally invasive surgery, as compared to laparotomy, resulted in less blood loss and less wound infections occured, but more operative time. Endoscopic approach evoked less urinary bladder dysfunction.
However, the most important finding is that all four groups of patients demonstrated similar 3-year survival outcomes. This very awaited study provides a sound basis to promote minimally invasive approach in the treatment of endometrial cancer, as opposed to widely discussed data from studies on laparoscopic treatment of cervical cancer.
I highly recommend publishing this paper in the present form to the interest of readers.
→Response: Thank you for the review and recommendation of publishing our article.